# Recent Achievements in Electrochemical and Surface Plasmon Resonance Aptasensors for Mycotoxins Detection

Gennady Evtugyn [1,2,*], Anna Porfireva [1], Tatjana Kulikova [1] and Tibor Hianik [3,*]

1   A.M. Butlerov' Chemistry Institute of Kazan Federal University, 18 Kremlevskaya Street, 420008 Kazan, Russia; Anna.Porfireva@kpfu.ru (A.P.); TaNKulikova@kpfu.ru (T.K.)
2   Analytical Chemistry Department of Chemical Technology Institute of Ural Federal University, 19 Mira Street, 620002 Ekaterinburg, Russia
3   Department of Nuclear Physics and Biophysics, Comenius University, Mlynska dolina F1, 84248 Bratislava, Slovakia
*   Correspondence: Gennady.Evtugyn@kpfu.ru (G.E.); Tibor.Hianik@fmph.uniba.sk (T.H.); Tel.: +7-843-2337491 (G.E.); +421-2-60295683 (T.H.)

**Abstract:** Mycotoxins are secondary metabolites of fungi that contaminate agriculture products. Their release in the environment can cause severe damage to human health. Aptasensors are compact analytical devices that are intended for the fast and reliable detection of various species able to specifically interact with aptamers attached to the transducer surface. In this review, assembly of electrochemical and surface plasmon resonance (SPR) aptasensors are considered with emphasis on the mechanism of signal generation. Moreover, the properties of mycotoxins and the aptamers selected for their recognition are briefly considered. The analytical performance of biosensors developed within last three years makes it possible to determine mycotoxin residues in water and agriculture/food products on the levels below their maximal admissible concentrations. Requirements for the development of sample treatment and future trends in aptasensors are also discussed.

**Keywords:** mycotoxin analysis; aptamer; aptasensor; electrochemical sensor; surface plasmon resonance

## 1. Introduction

Growing environmental pollution and severe climate change result in the extraordinary growth of risks related to chemical and biological hazards [1]. Pesticide residues, industrial chemicals, food processing by-products, etc., that are released into the environment not only exert direct adverse effect on human health but also have negative influence on natural processes and food chain transfer of toxic species [2–4]. Among many others, mycotoxins are frequently mentioned as one of most burning problems in food safety and human health control [5,6]. Mycotoxins are secondary metabolites of fungi (*Aspergillus*, *Fusarium*, *Penicillium*, *Claviceps,* and *Alternania* genus of plants) [7]. They affect a broad range of agricultural products and arise in conditions of high humidity and increased temperature, promoting the growth of molds. Most mycotoxins are very toxic. They damage plasma membranes, nucleic acids, affect protein synthesis, and increase cancer risk [8–11].

Most mycotoxins are currently determined by various types of chromatography coupled with mass spectroscopy or fluorescence detectors [12,13]. Being very sensitive and reliable in laboratory analysis, such equipment is less compatible with specific the requirements offered by field applications directed to the on-site detection of contaminants [14]. The concept of early warning devices assumes the possibility of the semi-quantitative determination of toxic species to be performed by technicians with rather simple measurement protocols and in real time [15].

Chemical sensors can be considered as an alternative to conventional analytical techniques that meet these requirements and offer information on the potential risks related to chemical and biological species in the environment [16].

Biosensors utilize specific interactions between the analyte and a molecular receptor of biological origin, or a derivative of the latter, that is immobilized directly on the surface of a transducer. In biosensors, enzymes, antibodies, nucleic acids, etc., are attached onto the transducer so that the interaction between the analyte and the biorecognition layer affects a physical-chemical property, which is transformed into a physical signal (typically an electric signal) by the transducer, allowing the selective quantitative or semi-quantitative detection of the analyte [17]. Biosensors show unique sensitivity and selectivity toward many contaminants [18]. However, in the case of mycotoxins, their performance is still limited by the availability of appropriate biochemical receptors. Indeed, mycotoxins can be determined by direct electrochemical oxidation [19] on the electrode or by inhibition of acetylcholinesterase [20] and urease [21]. However, the concentrations achieved by these approaches remain higher than the maximal admissible levels established for mycotoxins. Immunochemistry offers quite sensitive and selective detection, but it is mostly time and labor consuming and suffers from the insufficient stability of immunoreagents.

Aptamers (from Latin *aptus* (fit) and Greek *meros* (part)) are synthetic oligonucleotides selected from a random nucleotide library by the combination of combinatorial chemistry and affinity chromatography against target analyte [22,23]. They exert extraordinary analyte binding efficiency that is comparable to that of antigen–antibody interactions. For this reason, they have also been called "synthetic antibodies" [24]. Meanwhile, aptamers are more stable toward oxidation and hydrolysis than antibodies and can be easily modified by the implementation of the terminal functional groups that are necessary for their integration in biosensor assembly and for the covalent attachment of the labels producing specific signals. Although first aptamers have been designed for binding protein molecules, their application for the detection of small molecules became popular in the past decade due to the advantages of detection of such interactions as hapten binding with antibodies.

Recently, the application of biosensors utilizing aptamers (aptasensors) in mycotoxin analysis has been reviewed with emphasis to the signal measurement mode or individual mycotoxins detected [25–27]. This review is mainly focused on the application of aptasensors for mycotoxin determination in environmental samples and food that have been described in the period from 2018 to 2021.

## 2. General Characterization of Mycotoxins

Mycotoxins are secondary metabolites of fungi. The most frequently mentioned mycotoxins are produced by the *Aspergillus*, *Fusarium*, *Penicillium*, and *Claviceps* as well as the *Alternania* genus of plants [28]. High humidity and increased temperatures promote the biosynthesis of mycotoxins.

Approximately 400 compounds are currently known to belong to mycotoxins. Mycotoxins affect a broad range of agricultural products including cereals, cereal-based foods, dried fruits, wine, milk, coffee beans, cocoa and meat products [29,30]. They cause damage to the plasma membranes and nucleic acids (DNA, RNA) and influence the synthesis of proteins, which can seriously affect the health of individuals and cause cancer, especially that of the liver and kidney. Concerning the carcinogenic properties of the mycotoxins, the International Agency for Research on Cancer (IARC) classification involves the following groups:

**Group 1**—carcinogenic to humans;

**Group 2A**—probably carcinogenic to humans (limited evidence on humans but sufficient in animals);

**Group 2B**—possibly carcinogenic to humans (limited evidence to humans and not sufficient evidence to animals);

**Group 3**—not classifiable as to its carcinogenicity to humans;

**Group 4**—probably not carcinogenic to humans.

Mycotoxins are rather stable and cannot be destroyed by heating, irradiation, or treatment with organic solvents. They exert serious risks even after the sterilization of the milk and food products [31]. They can be included in a food chain by the consumption

of animals. The source, potential danger, and maximal admissible levels of common mycotoxins are listed in Table 1.

**Table 1.** Toxicity and maximal admissible levels of most important mycotoxins [32].

| Mycotoxin | Fungal Source | IARC Group | Contaminated Food | Maximal Admissible Levels (µg/kg) | |
|---|---|---|---|---|---|
| | | | | USA Food and Drug Administration | European Food Safety Authority |
| Aflatoxins (B1, B2, G1, G2) | *Aspergillus flavus* *Aspergillus parasiticus* | 1 | Wheat, maize, rice, peanut, pistachio, almond, hazelnut, ground nuts, tree nuts, figs, cottonseed | 20 | 4–10 for total 2–5 for B1 0.1 for B1 in baby food |
| Aflatoxin M1 | Metabolite of aflatoxin B1 | 2B | Milk and dairy products | 0.5 | 0.05 0.025 baby milk |
| Fumonisin B1, B2, B3 | *Fusarium verticillionides* *Fusarium proliferatum* | 2B | Maize, asparagus, corn-based food, white and yellow popcorn, sweet corn | 2000–4000 | 800–1000 200 baby food |
| Ochratoxin A | *Aspergillus ochraceus* *Penicillium verrucosum* *Aspergillus carbonarius* | 2B | Cereals, coffee, cocoa, wine, beer, dried fruits, grapes, pig kidney | Not set | 3–10, 0.5 baby food |
| Patulin | *Penicillium expansum* | 3 | Maize, asparagus, apple, pears, grapes, vegetables, cereals and cheese. | 50 | 25–50 10 baby food |
| Zearalenone | *Fusarium graminearum* *Fusarium culmorum* | 2A | Wheat, corn, barley, oats, sorghum and sesame seeds, hay and corn silage. | Not set | 50–100 20 baby food |
| Deoxynivalenol | *Fusarium graminearum* *Fusarium culmorum* | 3 | Corn, wheat, oats, barley, rice, grains, beer, animal's kidney and liver, milk, eggs | 1000 | 750–1250 200 baby food |
| Nivalenol | *Fusarium graminearum* *Fusarium culmorum* | 3 | Oats, barley, maize, wheat, bread and fine bakery wares, pasta, cereals | Not set | 1.2 |
| T-2 toxin | *Fusarium sporotrichioides* | 3 | Maize, wheat, corn gluten feed, corn, gluten meal, barley, bran | Not set | 0.012–0.043 |

Below, the most important mycotoxins are briefly characterized:

***Aflatoxins*** comprise about 20 fungal metabolites produced in animal feeds and various food commodities and is mainly caused by *Aspergillus parasiticus* (B aflatoxins) and *Aspergillus flavus* (B and G aflatoxins). Indices "B" and "G" indicate blue and green fluorescence produced by the appropriate species under UV irradiation applied in thin layer chromatography for their detection. Aflatoxins were first discovered in the 1960s in the United Kingdom, where more than 100,000 turkey poultry birds died due to ground nut meal infected with *A. flavus* [33]. Aflatoxins enter the food because of a breach of proper storage conditions. The chemical structures of aflatoxins are presented in Figure 1.

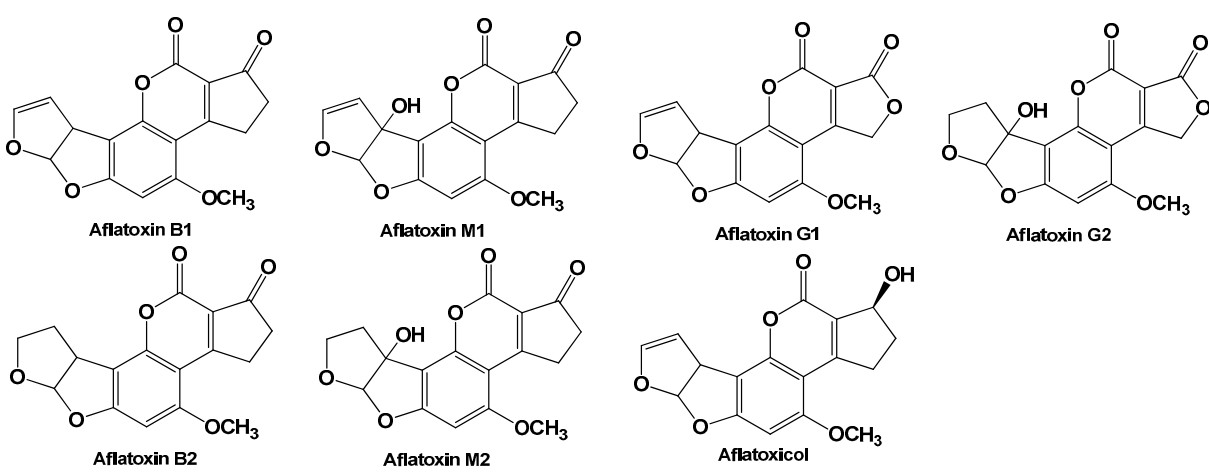

**Figure 1.** Chemical structure of aflatoxins.

In the case of acute aflatoxicosis, a large exposure can lead to the deaths of 25% of those who have been exposed. However, lethal cases are rather rare because people normally avoid consuming moldy products. Chronic poisoning, especially that of animals, is more frequent and can result in covalent DNA binding and the decreased production of proteins. This results in the decreased growth of animals and development of abnormalities. Cumulative aflatoxicosis results in a carcinogenesis [34], especially hepatocellular carcinoma, the sixth common world cancer [35].

Aflatoxin B1 (AFB1) and B2 are metabolically converted into aflatoxins M1 and M2, respectively. Their high solubility in lipids results in the accumulation of aflatoxins M1 and M2 in milk, which results in transfer to cheese and other dairy products. Aflatoxin M1 (AFM1) can be detected in animal tissues and fluids (urine and milk) at 12–24 h after the consumption of feed contaminated by AFB1. Aflatoxicol is a reductive metabolite of AFB1.

*Ochratoxin A* (OTA), first found in the Balkan region [36], is produced by *Aspergillus ochraceus* and *Penicillium verrucosum* sp. [37,38] (see chemical structure in Figure 2). In wine producing regions, *A. carbonarius* and other black-spored *Aspergillus,* section *Nigri* spp., are predominant OTA producers [39,40]. This is a polyketide mycotoxin that exerts nephrotoxic, immunosuppressive, teratogenic, carcinogenic, and cytotoxic properties. OTA contamination risk is mostly attributed to coffee, cereal grains, processed foods, beer, grapes, wine, cocoa, nuts, and dried fruits [41]. OTA exerts nephrotoxic, hepatotoxic, teratogenic, and immunotoxic effects on several species of animals and causes kidney and liver tumors in mice and rats [42–44]. OTA was assumed to be responsible for Balkan endemic nephropathy, a chronic tubulointerstitial kidney disease. It is also carcinogenic in the kidneys and liver (IARC Group 2B) [45,46].

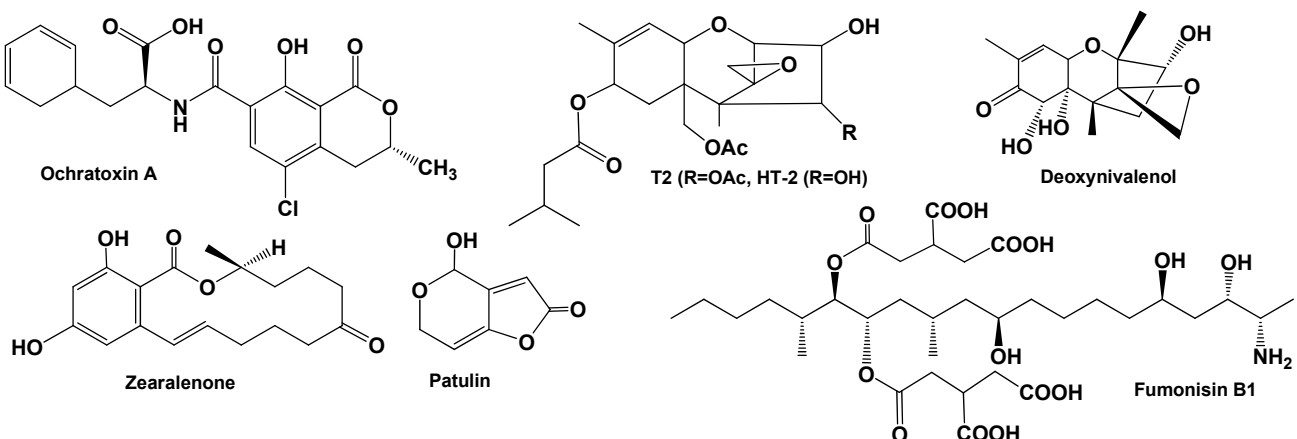

**Figure 2.** Chemical structure of ochratoxin A, trichothecene mycotoxins, patulin, fumonisin B1, and zearalenone.

*Fumonisins* are produced in cereals, usually by *Fusarium verticillioides, Fusarium proliferatum*, and other related species [47]. Their major influence on human health is caused by more than 15 fumonisin homologues which are mostly associated with contaminated maize, maize-based products as well as rice, wheat, barley, rye, oat, and grain [48]. Fumonisin B1 (Figure 2) is the most abundant. Together with fumonisin B2 and B3 (C-5 and C-10 dehydroxy analogs of fumonisin B1), they are the main food contaminants. The toxins have toxic effects on the liver and nephrons in tested animals [49]. Fumonisin B1 is implicated with hepatocarcinoma, immune system stimulation/suppression of the immune system, defects in the neural-tube, nephrotoxicity as well as other ailments. IARC characterized fumonisin B1 as belonging to group 2B [50].

*Patulin* is produced by some species of *Byssochlamys*, *Aspergillus*, and *Penicillium* [51]. Patulin was identified in agricultural crops, e.g., tomatoes, peppers, various fruits (pears, grapes, apples, and figs), seafood, ground cereals, and rice). High water and sugar content in fruits promote its synthesis [52]. Moreover, patulin has been detected in some manufactured dehydrated fruits, juices, and jams [53,54]. Acute exposure to patulin causes

gastrointestinal symptoms (nausea, vomiting, ulcers, intestinal hemorrhages etc.). In accordance with IARS, patulin is a Group 3 carcinogen. It is also linked to neurological, gastrointestinal, and immunological adverse effects [55].

*Trichothecene* mycotoxins are mainly produced by the *Fusarium* spp. and affect animals and humans through contaminated grains (wheat, oats, barley, maize, and rice) [56]. They are classified in accordance with their structure and producing organisms in four groups, i.e., type A (T2, HT2 toxins, neosolaniol, and diacetoxyscirpenol), type B (deoxynivalenol, nivalenol, 3-acetyldeoxynivalenol, and fusarenone), type C (crotoxin and baccharin), and type D (satratoxin G, H, roridin A, and verrucarin A). Frequently mentioned trichotocene mycotoxin structures are presented in Figure 2. Trichothecenes are commonly found on cereals grown in the temperate regions of Europe, America, and Asia (wheat, rye, barley, oats, and corn) [57,58]. They are absorbed via the gastrointestinal system [59] and cause refusal, immunological problems, vomiting, skin dermatitis, and hemorrhagic lesions [60,61]. Trichotocenes are also phytotoxic and can cause chlorosis, the inhibition of root elongation, and dwarfism [62]. Pigs and poultry are the most sensitive toward trichothecene mycotoxins.

*Zearalenone* (earlier F-2 toxin) (Figure 2) is produced by some *Fusarium* and *Gibberella* spp. Zearalenone and its hydroxylated derivatives are the only mycotoxins that exert a primary estrogenic effect due to its ability to bind to estrogen receptors [63]. Zearalenone causes infertility, abortion, or other breeding problems, especially in swine. Although the compound is not very toxic, 1–5 ppm is enough to poison swine directly and via a sow's milk. Zearalenone was found in moldy hay, high-moisture corn, and pelleted feed. The direct involvement of zearalenone in human toxicosis have not been confirmed, but it is considered as a potential hazard (endocrine disruptor) [64]. In 1993, it was included in Group 3 by the IARC (see above).

## 3. Conventional Methods of Mycotoxin Determination

Determination of mycotoxin contamination is obligatory in all areas, assuming the contact of the appropriate subjects with a human, especially through agriculture, food processing, food and feedstuffs. Analytical protocols approved for use in accredited laboratories involve mainly high-performance liquid chromatography (HPLC) with diode array, fluorescence, or mass detectors. Liquid chromatography–mass spectrometry (LC-MS) and gas chromatography–MS (GC–MS) instrumentations are routinely used for food and environmental contamination monitoring. Extraordinary selectivity and sensitivity and high throughput are mostly mentioned as their advantages [65]. The relatively low molar mass of mycotoxins has resulted in the preferable application of tandem mass spectroscopy coupled with HPLC [66,67]. Among ionization methods, electrospray ionization, matrix-assisted laser desorption ionization, electron ionization, and atmospheric pressure chemical ionization are commonly employed in mycotoxin detection [68].

The chromatographic determination of mycotoxin traces is commonly coupled with the enrichment and purification of the samples, which is performed by solid phase extraction. The use of immunochromatographic column suppressed background ionization in LC-MS detection. This increased the sensitivity of the mycotoxin quantification. A combination of various antibodies on sepharose column made the multiplex assay of mycotoxins in agriculture products possible [69]. The optimization of antibody immobilization and column reactivation allowed for multiple uses with no enrichment efficiency losses [70]. A similar approach can be employed with the columns derived with aptamers. Thus, AFB1 was determined by HPLC with fluorescent detection and photochemical derivatization of the analyte [71]. Aflatoxin enrichment and clean-up on the appropriate column containing the aptamer 5′-GTT GGG CAC GTG TTG TCT CTC TGT GTC TCG CGC CCT TCG CTA GGC CCA CA-3′ could be performed up to 20 times with the same column. A limit of detection (LOD) of 0.05 ng/mL and recoveries from 91.8 to 108.6% were reported for lotus seed analysis. One-pot synthesis of monolith of polyhedral oligomeric silsesquioxane modified with thiolated aptamer 5′-SH-C$_6$-GAT CGG GTG TGG GTG GCG TAA AGG

GAG CAT CGG ACA-3′ has been described for the determination of OTA [72]. HPLC with fluorescent detection and on-line preconcentration columns made it possible to determine the level of OTA as low as 0.025 ng/mL. More recently, aptamer assisted ultrafiltration cleanup with a fluorescence detector has been described for the HPLC determination of OTA in green coffee [73]. The 5′-GAT CGG GTG TGG GTG GCG TAA AGG GAG CAT CGG ACA-3′ aptamer was mixed with the OTA containing extract and then separated on the ultrafiltration filter. The LOD of 0.05 ng/mL and a recovery of 97.7% were reported for the spiked samples. An aptamer-functionalized capillary monolithic column was applied for patulin determination with a LOD of 2.17 pM and a linear range of concentration from 8.11 pM to 8.11 nM [74]. The thiolated aptamer 5′-SH-C$_6$-CAG CTC AGA AGC TTG ATC CCG GCC CGC CAA CCC GCA TCA TCT ACA TCA CTG ATA TTT TAC CTT GAC TAT CAG TCG TGC ATC TG-3′ was immobilized on the capillary monolithic column of Au nanoparticles-thionine-poly((glycidyl methacrylate)-co-poly (ethylene glycol) diacrylate)).

The preliminary screening of mycotoxin contamination can be performed using enzyme-linked immunosorbent assay (ELISA). High sensitivity of fluorescent [75], colorimetric [76–78] and chemiluminescent [79,80] signals is achieved by coupling enzymes and enzyme mimicking nanomaterials [81] with additional amplification, e.g., aggregation of Au nanoparticles [82], magnetic separation [83], or chromophore reactions [84]. Applicability of ELISA methods are often limited by a high matrix effect and the low stability of the antibodies and enzyme labels [85].

Being very sensitive, conventional analytical techniques offer strict requirements to the quality of auxiliary reagents and material qualification. The development of biosensors, and particularly the development of aptasensors, for mycotoxin determination is not aimed at substituting them, but instead allows for increasing the area of preliminary testing of environmental and agriculture samples to avoid human poisoning and other severe consequences of mycotoxin contamination for the environment.

## 4. Aptamers Utilized in Aptasensor Assembly

The appropriate protocol elaborated for aptamer selection is called SELEX (Systematic Evolution of Ligands by Exponential enrichment) [86,87]. In SELEX, a biological target is incubated with a pool of $10^{14}$~$10^{16}$ random oligonucleotides consisting typically of 40–100 bases. Target-binding oligonucleotides are then separated from the others using affinity chromatography. The bonded oligomers are then amplified by polymerase chain reaction (PCR). The protocol is repeated, and after several rounds, the most effective aptamers are enriched and then sequenced. Besides DNA libraries, RNA libraries have also been successfully used for SELEX [88,89]. However, RNA aptamers are sensitive toward RNAases and should be receive additional protection. Recent modifications of the SELEX protocol are aimed at its simplification and the modification of the produced aptamers in order to change the appropriate binding constants or selectivity toward chemically relative compounds [90]. Thus, in immunoprecipitation coupled SELEX (IP-SELEX), the complexes of target proteins with aptamers are separated by immunoprecipitation [91]. In capture SELEX, small molecules are bound to the middle part of the oligonucleotide sequence and the complex is immobilized by terminal items to magnetic beads [92,93].

In cell SELEX, live cells are applied for aptamer selection [94]. This method was developed for applications in oncology. SELEX protocols coupled with capillary electrophoresis (CE-SELEX) [95], microfluidics (M-SELEX) [96], and atomic force microscopy (AFM-SELEX) [97] are used for the acceleration of the rounds required for aptamer selection and operation with ultra-low volumes of reactants to cause the production to be cheaper. Thus, microfluidic chips can contain PCR thermocycler, pressurized reagent reservoir manifold, actuatable valves, and other items required for their operation on a single chip [98]. Specific properties of the SELEX based protocols are briefly summarized in Table 2.

**Table 2.** Key aspects, advantages, and disadvantages of the currently used SELEX methods [99].

| Method | Key Aspects | Advantages | Disadvantages |
|---|---|---|---|
| IP-SELEX | Includes immunoprecipitation. | Selects aptamers against proteins under normal physiological conditions. Increased affinity and specificity. | More time-consuming than standard SELEX. |
| Capture-SELEX | Oligonucleotide library immobilized on a support instead of the targets to identify aptamers against small molecules. | Suitable for the selection of aptamers against small molecules. Immobilization of the target not required. Used for the discovery of structure-switching aptamers. | Some oligonucleotides from the library might be not released/selected. |
| Cell-SELEX | Utilizes whole live cells as targets for selection of aptamers. | Prior knowledge of the target not required. Aptamers are selected against molecules in their native state. Many potential targets available on the cell surface. Protein purification is not required. | Suitable for cell surface targets. Requires high level of technical expertise. Costly. Time consuming. Post SELEX identification of the target is required. |
| CE-SELEX | Involves separation of ions based on electrophoretic mobility. | Fast. Only few (1–4) rounds of selection required. Reduced non-specific binding. Target immobilization is not required. | Not suitable for small molecules. Expensive equipment. |
| M-SELEX | Combines SELEX with a microfluidic system. | Rapid. Very efficient (only small amounts of reagents needed).Applicable to small molecules. | Low purity/recovery of aptamers. Target immobilization required. |
| AFM-SELEX | Employs AFM to create 3D image of the sample surface. | Able to isolate high affinity aptamers. Fast (only 3–4 rounds are required). | Expensive equipment is required. Immobilization of target and aptamers are required |

Efficiency of the aptamer-target binding is sensitive to the 3D structure of the aptamer molecule. For this reason, aptamers are often additionally modified with terminal linkers that separate its binding site from the carrier and diminish possible steric limitations of analyte access [100]. Homonuclear short sequences like $(dA)_n$-(dA-adenine deoxynucleotide) [101], polyethylene glycol chains [102], diacylglycerol [103], and cholesterol [104] based tails are used for this purpose. Some of the above modifications were first intended for the medical applications of aptamers, i.e., their implementation in vesicles or use in drug delivery systems. The chemical modification of the aptamers by the thiol group, biotin residue, amino, or carboxylic groups is mostly directed to their covalent immobilization on solid surfaces, including polymer carriers and the transducer interface.

Steric changes followed by analyte recognition are significantly higher in aptamers against antibodies. This results from the more flexible structure of aptamers and the high density of negative charge in the backbone of phosphate residues. The rearrangement of aptamer molecules has been successfully applied for the amplification of the signal of appropriate aptasensors. Two most frequently used approaches can be mentioned: (i) G4 quadruplexes and (ii) pinhole aptamers application.

Aptamers containing guanine rich domains can fold into a compact structure consisting of one or several flat tetramers (guanine quadruplexes) stabilized with a central metal ion (sodium in physiologically normal conditions) (Figure 3). The guanine quadruplex is composed of two guanine tetrads stabilized by Hogsten bonds. The formation of G4 quadruplexes results in the denser packing of aptamers on the solid interface and improves electrostatic interactions with the positively charged binding sites of the ligands. The latter one is explained by negative charge density of G4 quadruplexes that is twice that of linear DNA [105]. Van der Waals, $\pi$-$\pi$ stacking, and hydrophobic interactions also participate in such binding. Aptamer folding into a 3D structure requires a certain buffer composition and is controlled by pH, ionic strength, and temperature [106]. The analysis of the G4 quadruplex structure can be performed using software available on the Internet, e.g., QGRS Mapper [107] or UNAfold web server [108].

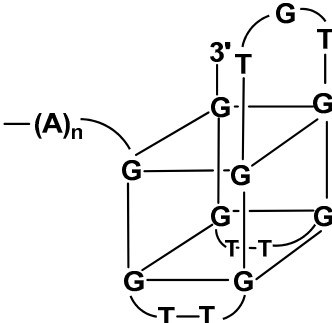

**Figure 3.** G4-guadruplex formed in guanine rich part of the aptamer molecule [109].

Pinhole, also named hairpin or stem-loop aptamers, contain pseudo-circle areas that are separated with self-hybridized double-stranded fragments. The structures selected from random libraries are presented in Figure 4 [110]. The number of unhybridized "circles" can vary from one to four. In a recognition event, some of the "circles" are opened to a linear configuration and are bonded to the analyte molecule [111]. This results in dramatic changes to the specific charge and volume of the aptamer molecule. Moreover, terminal groups introduced for electrochemical/optical detection change their relative position.

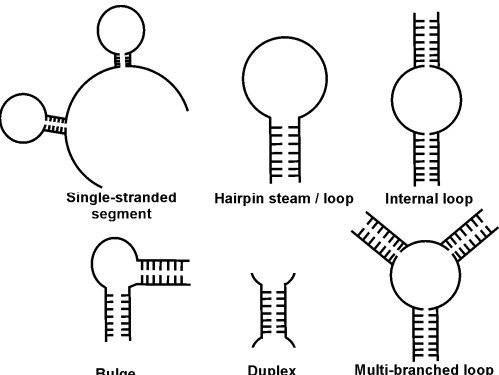

**Figure 4.** Pinhole structure of aptamers.

This offers new opportunities for analyte determination. Examples of the application of pinhole aptasensors are given below in the appropriate sections describing electrochemical and optical aptasensors. The design of the primary structure of such aptamers is intended to reach its most stable conformation after analyte binding and controlling the changes of spatial structure of the aptamer that are required for sensitive detection [112]. They are also successfully applied in optical sensors. Thus, the use of a fluorescent label and quencher on both sides of the loop makes it possible to obtain a switch-on aptasensor that produces irradiation in the analyte bonding, resulting in the separation of both labels from each other [113]. Similar changes following configuration changes of the aptamer loops can be obtained for displacement protocols [114] and for Förster resonance energy transfer [115,116]. The aptamer sequences designed for mycotoxin determination are presented in Table 3.

**Table 3.** The structure of the DNA aptamers selective toward mycotoxins (see also Section 3).

| Mycotoxin | Sequence 5′-3′ | Ref. |
|---|---|---|
| Aflatoxin B1 | GTT GGG CAC GTG TTG TCT CTC TGT GTC TCG TGC CCT TCG CTA GGC CCA CA | [106,117] |
| | AGC AGC ACA GAG GTC AGA TGG TGC TAT CAT GCG CTC AAT GGG AGA CTT TAG CTG CCC CCA CCT ATG CGT GCT ACC GTG AA | [118] |
| Aflatoxin M1 | ACT GCT AGA GAT TTT CCA CAT | [119] |
| | GTT GGG CAC GTG TTG TCT CTC TGT GTC TCG TGC CCT TCG CTA GGC CCA CA | [120] |
| | ATC CGT CAC ACC TGC TCT GAC GCT GGG GTC GAC CCG GAG AAA TGC ATT CCC CTG TGG TGT TGG CTC CCG TAT | [121] |
| Fumonisin B1 | CGA TCT GGA TAT TAT TTT TGA TAC CCC TTT GGG GAG ACA T | [122] |
| | ATA CCA GCT TAT TCA ATT AAT CGC ATT ACC TTA TAC CAG CTT ATT CAA TTA CGT CTG CAC ATA CCA GCT TAT TCA ATT AGA TAG TAA GTG CAA TCT | [123] |
| | ATA CCA GCT TAT TCA ATT AAT CGC ATT ACC TTA TAC CAG CTT ATT CAA TTA CGT CTG CAC ATA CCA GCT TAT TCA ATT | [124] |
| | AAT CGC ATT ACC TTA TAC CAG CTT ATT CAA TTA CGT CTG CAC ATA CCA GCT TAT TCA ATT | [125] |
| Ochratoxin A | GAT CGG GTG TGG GTG GCG TAA AGG GAG CAT CGG ACA; TGG TGG CTG TAG GTC AGC ATC TGA TCG GGT GTG GGT GGC GTA AAG GAG CAT CGG ACA ACG | [126] |
| Patulin | GGC CCG CCA ACC CGC ATC ATC TAC ACT GAT ATT TTA CCT T | [127] |
| | SH-CAGCTCAGAAGCTTGATCCT-GGCC CGC CAA CCC GCA TCA TCT ACA CTG ATA TTT TAC CTT GAC TCG AAG TCG TGC ATC TG | [128] |
| T-2 Toxin | GTA TAT CAA GCA TCG CGT GTT TAC ACA TGC GAG AGG TGA A | [129] |
| Zearalenone | TCA TCT ATC TAT GGT ACA TTA CTA TCT GTA ATG TGA TAT | [130] |

Regarding the comparison of antibodies and aptamers, higher thermal and chemical stability are mostly mentioned for aptamers utilized in appropriate biosensors [109]. Although the use of nanobodies [131] and Fab antibody fragments [132] compensates for the difference in the molecular weight and size of the receptors, the site specific immobilization of aptamers offers higher density positioning on the support and higher sensitivity of the appropriate signals toward an analyte binding. Direct comparison of both receptors has been presented for the optical detection of analyte binding in [133]. Nevertheless, in some works devoted to sandwich immunoassay, aptamers were used instead of primary/secondary antibodies because of a less steric binding hinderance and a higher sensitivity of label detection. Thus, in cardiac infarction biomarker detection, the use of aptamers increased the current response toward the analyte in the picomolar range of its concentration by two [134]. In sulfamethazine detection, signal enhancement was related to the accumulation of a redox indicator, Methylene blue (MB), after the formation of antibody–analyte–aptamer complex on the surface of an electrode modified with a carboxylated g-$C_3N_4$ layer [135]. In the future, a similar approach might be interesting in mycotoxin determination as well.

## 5. Aptasensing Strategies

### 5.1. Immobiization of Aptamers

Aptamers are mostly attached to the solid interface to separate the analyte bonded in specific complex and simplify the detection of labels present in the aptamer structure. In addition to the signal record requirements, immobilization makes aptamers more stable in the dry storage period and accelerates the preparation of the appropriate aptasensor to measurement conditions [136]. All the protocols of aptamer immobilization can be subdivided into three groups in accordance with the driving forces of the process.

*Physical immobilization* is based on multiple weak interactions between the solid support or immobilization matrix and aptamer molecule. For aptasensor development, electrostatic interactions are of the most importance though hydrophobic interactions, and hydrogen bonds are also considered for some carriers [137]. In physical immobilization, an aptamer retains its favorite structure for interaction with the analyte molecule. Meanwhile, physical immobilization is reversible, and the aptamer can be released from the immobilization state with sharp environment changes (pH shift, dramatic changes of the ionic

strength, etc.) The reversible entrapment–release of aptamer molecules can be used for the renewal of the aptasensor interface.

*Covalent immobilization* assumes the formation of one or several covalent bonds between the terminal groups of the aptamer molecule and the carrier. This reaction results in the formation of a very stable product that can be stored for a longer period against physical immobilization. Meanwhile, the structure of the aptamer can vary from that of native molecule, and this can affect the efficiency of target interactions with analyte molecules. Some typical schemes of covalent immobilization are outlined in Figure 5.

**Bridging via glutaraldehyde reaction**

**Carbodiimide binding**

**Aryldiazonium salt grafting**

**Figure 5.** Glutaraldehyde cross-binding of aminated aptamer and carrier. EDC—1-ethyl-3-(3-dimethylaminopropyl)-carbodiimide, NHS—*n*-hydroxysuccinimide.

They involve glutaraldehyde cross-linking [138], carbodiimide [139], and arylazide [140, 141] based binding. It should be noted that contrary to physical immobilization, covalent binding requires the modification of the primary aptamer sequence and the introduction of certain chemically active groups in the carrier or transducer surface. The immobilization of the thiolated aptamers to Au also refers to covalent immobilization due to the spontaneous formation of Au-S bonds [142].

*Miscellaneous methods of immobilization* combine the principles and advantages of both physical and covalent immobilization. In all of them, aptamers are directly modified by the introduction of certain reactive groups required for the following assembling of the products. The last stage of the process does not assume the covalent binding of the aptamer molecule.

The described immobilization protocols are outlined in Figure 6.

Avidin (streptavidin or neutravidin)–biotin binding is one of the most frequently used examples. High efficiency of the appropriate interaction (dissociation constant $K_D \sim 10^{-14}$–$10^{-15}$ M [143]) as well as the ability to bind up to four biotin residues makes such pairs very effective in the assembling of regular layers of aptamers either for immobilization or for label attachment. The latter application is stimulated by the availability of commercial preparations for biotinylated enzymes and by the possibility to construct universal sensing systems with different aptamers but with the same labeling and signal detection approach. Consecutive interaction of an aptamer with the analyte molecule followed by label addition minimizes steric limitations of interactions and improves the analytical performance of the appropriate aptasensors.

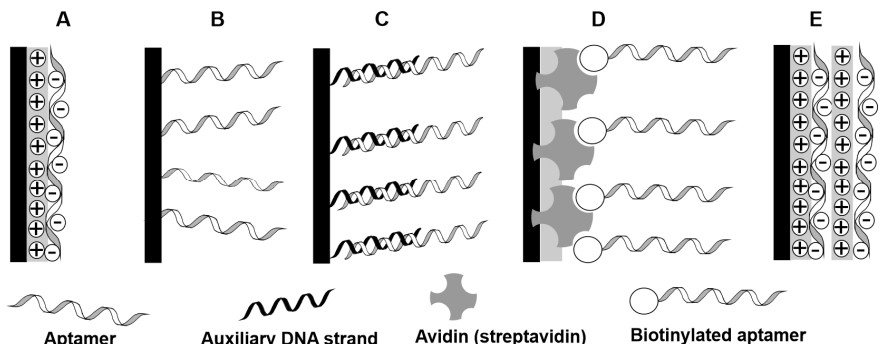

**Figure 6.** Common immobilization protocols applied for aptamer immobilization. (**A**) electrostatic immobilization on charged support; (**B**) covalent attachment via terminal functional group; (**C**) Affinity immobilization by hybridization with auxiliary single-stranded DNA sequence; (**D**) immobilization via avidin (streptavidin)—biotin binding; (**E**) inclusion in polyelectrolyte layers or ordered structured by preferably non-covalent interactions.

Hybridization of an aptamer with an auxiliary DNA sequence covalently attached to the carrier/transducer is another miscellaneous method described as affinity inhibition. Here, the immobilization efficiency is controlled by a primary sequence of DNA nucleotides that are complementary to an appropriate piece of the aptamer structure [144]. As a result, a double-stranded DNA fragment is formed and separates the aptamer and transducer interface. This might be important for the detection of bulky analyte molecules due to the suppression on steric factors and the pre-organization of a binding site on the carrier surface. Such an immobilization protocol offers additional opportunities for signal generation. Thus, the reaction with an analyte can compete with the interaction of complementary strands so that the aptamer leaves the transducer in the target binding. If the ds-DNA fragment is not involved in binding the analyte molecule, the hybridization seems to be very similar to conventional immobilization via the terminal functional group described above. In case of pinhole aptamers, such hybridization is not involved in the immobilization of the aptamer commonly based on terminal group binding. However, it requires the fixing of the circle configuration of the loop. Although hybridization is strong enough to maintain such configurations, the reverse process can be rather easily achieved through treatment with a concentrated electrolyte solution or pH shift.

Immobilized aptamers are often placed in a hydrophilic environment to stabilize their structure and ability to bind analyte molecules. Hydrophilic media can be a part of immobilization media if the aptamer is entrapped in a polymeric layer or interacts with polyelectrolytes, e.g., poly (ethylene imine) (PEI) [145,146]. They can also be a part of signal generation system in case of redox active polymers and electropolymerization products. Thus, cationic polyelectrolytes affect the aggregation of metal nanoparticles in the colorimetric detection of aptamer–analyte interactions [147] and act as the molecular "glue" in the assembly of aptamer containing polyelectrolyte complexes [148].

*5.2. Assembling of Electrochemical Aptasensors*

The immobilization of aptamers is not the only requirement for aptasensor assembling. Aptasensor design assumes the possibility to target interaction on the transducer interface and its conversion in the signal. In the case of electrochemical sensors, analyte binding should change either the current related to the redox conversion of special species (labels or redox indicators) or the permeability of the surface layer toward small ions. All the variety of measurement principles can be divided in two large groups, i.e., label-free and label-based techniques.

In label-free techniques, aptamers change their conformation in a recognition event from a linear one to a G4 quadruplex or other structures. This results in significant consolidation of the surface layer. All the reaction participants (aptamer and analyte) are electrochemically inactive, so the permeability of the surface layer is significantly in-

creased. This can be quantified using diffusionally free redox indicators. In most works, the $[Fe(CN)_6]^{3-/4-}$ redox indicator is applied for this purpose. Its response toward conformational changes is sensitive because of the electrostatic repulsion from the negatively charged aptamer chain phosphate residues. As a result of target interaction, peak currents related to the indicator progressively decrease with an increase in analyte concentration. Both direct current cyclic voltammetry and differential pulse voltammetry (DPV) are used in such measurements. Alternatively, the same indicator is applied in faradaic electrochemical impedance spectroscopy (EIS). In this method, the electrode is polarized at the equilibrium potential determined as a half-sum of the peak potentials of $[Fe(CN)_6]^{3-/4-}$. An alternative voltage of small amplitude (approximately 5 mV) is then applied and impedance is assessed from the alternate current changes. Charge transfer resistance depends both on the diffusion coefficient and electrostatic interaction on the electrode interface so that even small analyte molecules can be effectively determined with this method. Comparing to voltammetric techniques, EIS provides the determination of lower concentrations but is also more sensitive toward unspecific adsorption on the electrode interface, especially in the case of real sample assay. If Au electrodes are used, unspecific adsorption can be suppressed by covering free parts of the electrode surface with monolayers of thiols, e.g., thiohexanol [149] or alkylmercaptans [150]. Bovine serum albumin commonly used in immunoassay techniques for the same superpose is less effective. The analytical performance of label free aptasensors can be additionally improved with the implementation of bulky charged groups, e.g., poly(aminoamine) dendrimers [117], that affect electrostatic interaction and additionally prevent the transfer of the redox indicator to the electrode. The application of $[Fe(CN)_6]^{3-/4-}$ has some limitations caused by pH and interference with the anodic dissolution of Au, but they are not so important in most aptamer assemblies.

Labels that differ from redox indicators because of covalent bonding to the aptamer chain or auxiliary DNA strands can also be applied for signal generation. Their signals are recorded by DPV or square wave voltammetry (SWV) are more stable than those of indicators because there are no redox center losses during the aptasensor treatment (washing, reagent addition, sample incubation etc.). Labels are selected in accordance with their redox potential, chemical stability, and the reversibility of redox conversion [151]. It is also desirable that the redox response of a label is pH independent. Although many different labels have been described in aptasensor assembly in the past decade, recent works on mycotoxin determination describe only few of them, i.e., ferrocene, methylene blue, thionine and neutral red. In some cases, redox active polymers applied for aptamer immobilization also serve as redox labels. Changes in the label signals are more variable than those of redox indicators. If the signal decays with increased analyte concentration, aptamers are called as signal-off (switch-off) aptasensors. In the opposite case, signal-on (switch-on) aptasensors are considered.

Table 4 summarizes the information on the sensing layer assembly, signal generation mode and analytical characteristics of mycotoxin determination with electrochemical aptasensors during 2018–2021. Similar data within the previous period can be found in reviews [108,152–154].

**Table 4.** Electrochemical aptasensors for mycotoxin determination (2018–2021).

| Transducer | Transduction Principles | Samples Analyzed | LOD, Linearity Range | Ref. |
|---|---|---|---|---|
| | | Aflatoxin B1 | | |
| Glassy carbon electrode (GCE) covered with reduced graphene oxide (rGO) polyaniline/nanoAu/MoS$_2$ composite | Differential pulse voltammetry (DPV), electrochemical impedance spectroscopy (EIS) with $[Fe(CN)_6]^{3-/4-}$ redox indicator. | Wine | LOD 0.003 fg/mL, 0.01–1.0 fg/mL (DPV) | [155] |
| Indium-tin oxide (ITO) electrode covered with nanoAu/polyaniline | EIS with $[Fe(CN)_6]^{3-/4-}$ redox indicator. | Corn | LOD 0.05 ng/mL, 0.1–100 ng/mL | [156] |
| Au electrode with immobilized tetrahedral DNAs bearing auxiliary DNA sequences complementary to aptamers. | Mycotoxin binding releases aptamers from the surface, auxiliary DNA binds to complementary sequences bearing Au nanoparticles modified with peroxidase. Enzyme activity measured by redox current thionine utilized as a substrate | Rice, wheat powder | LOD 0.01 fg/mL, 0.1 fg/mL–0.1 μg/mL | [157] |
| Screen-printed electrode modified with magnetically collected Fe$_3$O$_4$@Au nanoparticles with aptamer immobilized via Au-SH bonds | EIS with $[Fe(CN)_6]^{3-/4-}$ redox indicator. | Peanut | LOD 15 pg/mL, 20 pg/mL–50 ng/mL | [158] |
| GCE modified with Au nanoparticles and β-cyclodextrin. | Aptamer is first hybridized with complementary DNA sequence with terminal ferrocene label. Mycotoxin binding releases auxiliary DNA. Ferrocene group is involved in the inclusion complex with the macrocycle; charge transfer resistance increases. DPV signal of ferrocene increases with the analyte concentration. | Peanut oil | EIS: LOD 0.049 ng/mL (0.147 pM), 0.1–10 ng/mL | [159] |
| GCE modified with rGO-thionine composite followed by electrodeposition of Au nanoparticles and immobilization of auxiliary DNA sequence complementary to aptamer bearing ferrocene label | Analyte binding results in release of aptamer form the electrode interface. Rational signal measurement based on simultaneous monitoring of ferrocene and thionine signals with alternating current voltammetry. | Peanut | LOD 0.016 ng/mL, 0.05–20 ng/mL | [160] |
| Au electrode modified with thiolated stem-loop aptamer with methylene blue on the opposite end | Square wave voltammetry (SWV) signal of methylene blue changing with target reaction resulted in transformation of the initial aptamer structure. Reaction is amplified by addition of short DNA sequence complementary to the aptamer. | Beer, white wine | LOD 8 nM, 8 nM–4 μM | [161] |

**Table 4.** *Cont.*

| Transducer | Transduction Principles | Samples Analyzed | LOD, Linearity Range | Ref. |
|---|---|---|---|---|
| Screen-printed electrode with electropolymerized poly(aniline-anthranilic acid) film and covalently attached BSA-aflatoxin conjugate | Reaction with biotinylated aptamer is followed by attachment of streptavidin-alkaline phosphatase conjugate. DPV detection of enzyme activity via redox current of 1-naphotl formed from 1-naphtylphosphate as enzyme substrate. | Maize flour | LOD 0.086 ng/mL, 0.1–10 ng/mL | [162] |
| Au electrode modified with thiolated stem-loop aptamer bearing methylene blue at internal thymine fragment | Reaction with mycotoxin makes methylene blue available for electron transfer measured in SWV mode. | Wine, milk, and corn flour | LOD 6 pM | [163] |
| Aflatoxin M1 | | | | |
| Poly(neutral red) with carboxylated pillar[5]arene bearing monomeric dye and aminated aptamer | EIS with $[Fe(CN)_6]^{3-/4-}$ redox indicator | Milk and milk products | LOD 0.5 ng/mL, 5–120 ng/L | [164] |
| Au modified with self-assembled layer of *n*-doped graphene nanosheets and carboxylated polystyrene nanospheres followed by carbodiimide binding of aminated aptamer | EIS with $[Fe(CN)_6]^{3-/4-}$ redox indicator | Oil, soy sauce | LOD 2 pg/mL, 0.01–10 ng/mL | [165] |
| Hairpin shaped aptamer with Au nanoparticles and complementary strand immobilized on golden screen-printed electrode | DPV of diffusionally free methylene blue added after the analyte incubation | Human blood serum, milk | LOD 0.9 ng/L, 2–600 ng/L | [166] |
| Deoxynivalenol | | | | |
| Iron nanoflorets graphene nickel foam as electrode, aptamer covalently attached via glutaraldehyde linking | Changes in the electric conductivity monitored by polarization curves | Plant extracts | LOD 2.11 pg/mL, 1 fg/mL–1 ng/mL | [167] |
| Ochratoxin A | | | | |
| GCE modified with Au nanoparticles with attached aptamer, sandwich protocol with Cd containing MOF particles as labels | DPV signal of Cd (II) ions in the structure of the label measured without its dissolution | Red wine | LOD 10 pg/mL, 0.05–100 ng/mL | [168] |
| Screen-printed carbon electrode covered with polythiophene-carboxylic acid with covalently attached aptamer | EIS with $[Fe(CN)_6]^{3-/4-}$ redox indicator | Coffee | LOD 0.125 ng/mL, 0.125–20.0 ng/mL | [169] |

**Table 4.** *Cont.*

| Transducer | Transduction Principles | Samples Analyzed | LOD, Linearity Range | Ref. |
|---|---|---|---|---|
| Au electrode covered with β-cyclodextrin onto MoS$_2$.nanoAu layer; aptamer is attached to the surface via supramolecular interaction with terminal Methylene blue group | Target interaction removes aptamer from the cyclodextrin moiety. Instead, ferrocene carboxylic acid is captured. The DPV signals of both methylene blue and ferrocene change synchronously. | Wine | LOD 0.06 nM, 0.1–50 nM | [170] |
| Au electrode covered with thiolated aptamer | Target interaction prevents aptamer cleavage caused by exonuclease enzyme; signal is enhanced by silver metallization of aptamer molecule. Ag oxidation DPV signal. | Beer | LOD 0.7 pg/mL, 1 pg/mL–0.1 μg/mL | [171] |
| Pencil graphite electrode electrografted with 4-amionobenzoic acid followed by covalent immobilization of aminated aptamer | EIS with $[Fe(CN)_6]^{3-/4-}$ redox indicator | Beer | LOD 0.1 ng/mL, 0.1–2.0 ng/mL | [172] |
| GCE covered with nitrogen doped graphene and saturated with Methylene blue | Aptamer hybridized with complementary DNA strand reacts with OTA, released DNA is adsorbed on the electrode and increases redox signal of methylene blue measured by SWV. | - | LOD 0.71 fg/mL, 1 fg/mL–1 μg/mL | [173] |
| Au electrode with covalently attached thiolated auxiliary DNA sequence complementary to aptamer | Signaling DNA probe bears Au nanoparticles and ferrocene label. Displacement protocol with DPV detection of ferrocene signal. | Wine | LOD 0.001 ppb, 0.001–500 ppb | [174] |
| Dual mode paper-based sensor. Aptamers were immobilized on chitosan functionalized MoS$_2$–Au@Pt. | CV, EIS. Catalyzed reduction of H$_2$O$_2$. | Corn | LOD 0.025 pg/mL, 0.0001–200 ng/mL | [175] |
| Au electrode modified with copolymer of pyrrole and pyrrole-3-qcetic acid followed by covalent binding of PAMAM G4 dendrimer and cross-lining of aptamer with glutaraldehyde | EIS with $[Fe(CN)_6]^{3-/4-}$ redox indicator | Wine | LOD 2 ng/mL, 2–6000 ng/mL | [176] |

**Table 4.** *Cont.*

| Transducer | Transduction Principles | Samples Analyzed | LOD, Linearity Range | Ref. |
|---|---|---|---|---|
| | | Patulin | | |
| Glassy carbon electrode modified with ZnO nanorods and Au nanoparticles | DPV signal of $[Fe(CN)_6]^{3-/4-}$ redox indicator | Apple juice | LOD 0.25 pg/mL, 0.50 pg/mL–50 ng/mL | [125] |
| Screen-printed carbon electrode "activated" by chemical grafting with diazonium salt, aminated aptamer with long PEG linker | EIS with $[Fe(CN)_6]^{3-/4-}$ redox indicator | Apple juice | LOD 1.25 ng/mL, 1–25 ng/mL | [177] |
| | | T2 toxin | | |
| GCE covered with polyaniline-MoS$_2$-chitosan-Au nanocomposite and thiolated aptamer. | GO-tetraethylene pentaamine–gold@platinum nanorods bearing auxiliary DNA complementary to aptamer are added together with analyte solution. Left free aptamer molecules form hybridization product amperometrically detected by electrocatalytic oxidation of hydrogen peroxide. | Canned beer | LOD 1.79 fg/mL, 10 fg/mL–100 ng/mL | [178] |
| | | Zearalenone | | |
| GCE modified with chitosan, acetylene black and multiwalled carbon nanotubes followed by Au deposition and covalent attachment of thiolated DNA sequence complementary to aptamer | Aptamer is covalently attached to carboxylated rGO nanoflackes. Its reaction with mycotoxin prevents binding to the electrode. In the opposite way, hybridization results in a sharp decrease of the surface layer permeability detected with EIS by ferricyanide redox probe. | Corn oil and corn flour | LOD 3.64 fg/mL, 10 fg/mL–10 ng/mL | [179] |
| Au electrode with covalently attached zearalenone conjugate | Indirect competitive assay with SWV or EIS measurements of permeability of the surface layer in the presence of $[Fe(CN)_6]^{3-/4-}$ redox indicator. | Maize grain | LOD 0.017 ng/mL, 0.01–1000 ng/mL | [180] |
| | | Fumonisin B1 and zearalenone | | |
| GCE covered with co-reduced MoS$_2$ and Au followed by covalent immobilization of thiolated aptamers against zearalenone and fumonisin B1 | Au nanoparticles modified with DNA sequences complementary to aptamers and saturated with thionine or 6-ferrocenelhexanthiol. Analyte binding resulted in release of the labels and changes in the signals of ferrocene and thionine recorded simultaneously in DPV mode. | Maize | Zearalenone: LOD 0.5 pg/mL, 0.001–10 ng/mL Fumonisin B1: LOD 0.5 pg/mL, 0.001–100 ng/mL | [181] |

DPV—differential pulse voltammetry, EIS—electrochemical impedance spectroscopy, LOD—limit of detection, rGO—reduced graphene oxide, CV—cyclic voltammetry, SWV—square wave voltammetry, MOF—metal-organic frameworks.

MB is one of the most interesting redox active components of aptasensors. Previously, this dye was applied for the detection of hybridization events with electrochemical DNA sensors. It could adsorb on the surface of single-stranded DNA molecules in the area of minor grooves [182] and simultaneously intercalate double stranded DNA as a hybridization product [183].

Later, double-stranded DNA attached to the Au surface via terminal functional group showed the possibility of the long-distance electron transfer along the chain of nucleobase pairs alternating with flat MB molecules [184]. Moreover, terminal MB molecules covalently bonded to DNA strands show excellent voltammetric response [185,186]. They can serve for the highly sensitive detection of both DNA specific interactions and aptamer–analyte binding. Thus, the use of a pinhole aptamer structure with terminal MB label made it possible to monitor of the Aflatoxin B1 binding by changes in the DPV signal of the label. For this purpose, the analyte was first added to the single stranded DNA complementary to the aptamer. In the absence of the target, hybridization elongated the distance from the dye molecule to the electrode. However, aflatoxin B1 stabilized loop configuration, so the signal of MB remained rather high [161].

To some extent, the same signal measure protocols are applicable to thionine. Additionally, it was used in competitive schemes where the first surface layer was saturated with the redox indicator and was then partially pushed out by competitive adsorption of the DNA molecules released in a specific aptamer interaction [160].

Contrary to them, ferrocene (Fc) is mostly used as labels due to the low solubility of free Fc in water. Fc and MB can be used together in so-called ratiometric DNA sensors [187]. Here, one label is used for immobilization control, and another one is attached to the opposite end of the DNA sequence and increases its signal after an analyte binding [170]. The use of auxiliary DNA hybridized to aptamers makes it possible to use the MB signal as a measure of aptamer release and the concentration of the analyte bonded to free aptamers [166]. An alternative approach assumes labeling different aptamers with the labels and detection of several mycotoxins in a single measurement. The only example within last three years describes determination of fumonisin B1 and zearalenone [181] with thionine and Fc, but the approach to multiplex assay is very promising, and more multisensors can be expected in the future. Principal schemes of label based aptasensors applicable for mycotoxin determination are outlined in Figure 7.

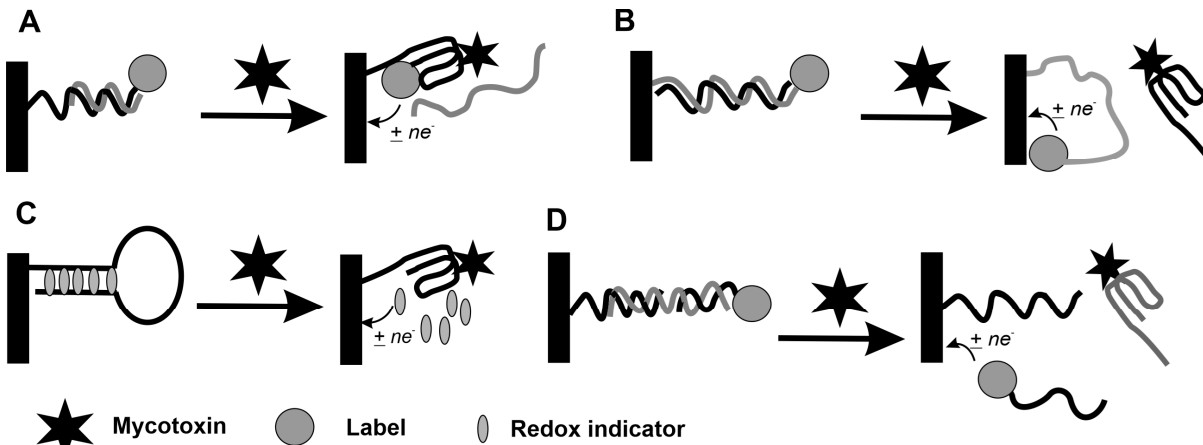

**Figure 7.** Schematic outlines of the signal generation modes in electrochemical aptasensors. (**A**) aptamer is attached to the electrode. Its folding releases auxiliary DNA and makes the distance of electron transfer narrower; (**B**) Auxiliary DNA is labeled and forms double-stranded helix with aptamer. Target reaction removes the aptamer and accelerates the electron transfer from the terminal label; (**C**) pinhole aptamer is saturated with redox active intercalator. Target reaction releases intercalator from the stem of the aptamer; (**D**) aptamer binds together two auxiliary DNA sequences. Its removal by binding with analyte releases labeled fragment producing redox signal.

### 5.3. Surface Plasmon Resonance Aptasensors

Optical methods are widely used in biosensing applications. Among them, the surface plasmon resonance (SPR), colorimetry, fluorescence spectroscopy, surface enhanced Raman spectroscopy (SERS), and electrochemiluminiscence (ECL) are effective for the detection of mycotoxins using nucleic acid aptamers as receptors. In this part, we focus on the SPR based aptasensors that are of substantial interest for practical applications in monitoring food safety. The interest in this technique is also supported by the appearance of commercial SPR devices [188–194], some of which allow field application as well as the use of appropriate SPR chips with well reproduced characteristics. Achievements in other optical biosensors for mycotoxin detection can be found in several recent comprehensive reviews [25,188–194].

SPR is a very effective label-free method to detect affinity interaction at surfaces. The transduction element consists of a glass prism with a thin metallic layer sputtered at its top. Gold is mostly used for this purpose. The gold layer serves to immobilize receptors such as aptamers or antibodies. In the measurement phase, a laser beam passes through a prism covered with a thin Au layer. Under total reflection conditions, the interaction between the free oscillating electrons of the metal and the light photons results in electron resonance. The generated evanescent electromagnetic field at the metal interface penetrates the neighboring dielectric medium. As a result, any changes on the SPR chip surface affect the conditions of the plasmon resonance. The described changes are quantified by variation in the total reflectance angle (Figure 8). The resonant angle plot changes or the light intensity as a function of analyte concentration serves as a calibration curve [195].

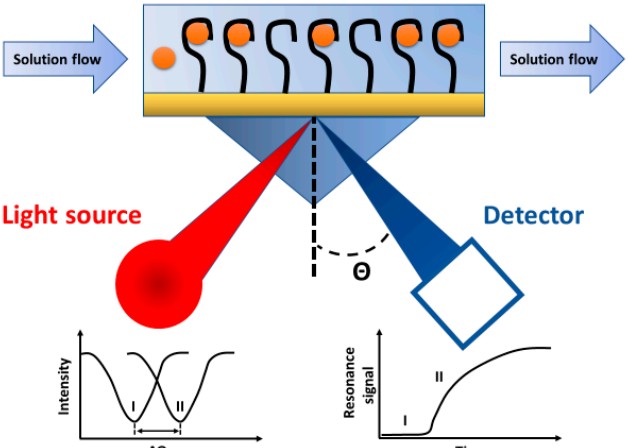

**Figure 8.** The scheme of the SPR set-up. Aptamers are immobilized at the gold layer sputtered on a glass prism. The binding of the analyte with aptamers resulted in the shift of the resonant angle. Adopted from [184] with permission of Elsevier.

SPR is traditionally applied for the detection of sufficiently large molecules with a molecular weight above 10 kDa that affect the dielectric properties of the interface [195]. Application of the method for small molecules assay usually requires amplification strategies, e.g., labeling analytes with appropriate species including nanoparticles, their involvement in the complexes, sandwich, or competitive inhibition assay [196].

One of the first SPR aptasensors for mycotoxins published by Zhu et al. [197] was focused on OTA detection. In this work, biotinylated DNA aptamers were immobilized at the streptavidin layer attached to the dextran matrix of the sensor chip by amine coupling. A rather low detection limit (0.005 ng/mL) was obtained, with the linear range being from 0.094 to 10 ng/mL. The aptasensor has been validated by measurements in spiked wine and peanut oil samples, where recoveries from 86.9% to 116.5% were obtained.

OTA detection coupled with QCM control of the surface layer assembly was reported in [198]. Immobilization of the thiolated aptamer was performed using a self-assembled mercaptoundecanoic acid monolayer by carbodiimide binding. The immobilization effi-

ciency was assessed by comparing the adsorption of $Ca^{2+}$ ions, which was found to be $1.42 \times 10^{14}$ molecules/cm$^2$. The SPR sensor in the measurement mode phase made it possible to detect OTA with a LOD of 5 pg/mL.

Further, the SPR aptasensor for AFB1 detection was developed by Sun et al. [199]. In this work, biotinylated aptamers were immobilized at the streptavidin coated SPR chip covered by dextran. The streptavidin had been bounded to the dextran by the amine coupling method. The addition of AFB1 resulted in an increase of the SPR signal (LOD 0.4 nM). Application of sandwich or competitive assay can increase sensitivity of mycotoxin detection. The **sandwich assay** is commonly performed in two different steps based on different affinity interactions. First, mycotoxin was added to the SPR sensor and allowed to interact with an aptamer/antibody immobilized on its surface. In the second step, another bioreceptor was added for signal amplification. This molecule should also interact specifically with mycotoxin accumulated on the chip surface and can bear gold nanoparticles able to enhance the SPR signal [200].

In **competitive assay**, the receptor (aptamer or antibody) is immobilized at the SPR chip surface. Prior to addition to the SPR aptasensor, a sample is mixed with the solution containing a target mycotoxin conjugate and bovine serum albumin (BSA) or mycotoxin derivative. Both mycotoxin and its conjugate/derivative interact with the same aptamer binding site so that the resulting signal will be inversely proportional to the mycotoxin concentration [201]. The advantage of the SPR over conventional optical sensors is that it can be used for measurements even in non-transparent liquids such as food samples or milk.

As an example, this approach was used for the detection of AFB1 in vinegar. The SPR chips were first modified with carboxymethylated dextran bearing streptavidin molecules covalently attached to the carrier. The biotinylated aptamer was specified as specific receptor. The described SPR sensor made it possible to detect AFB1 with a LOD of 0.19 ng/mL in the linear range of concentrations from 1.5 to 50 ng/mL.

The sensor has been validated in spiked samples of vinegar with recovery in a range 96.3–117.8% [202]. The combination of SPR and spectroscopic ellipsometry has been also applied for mycotoxin detection. This method is known as total internal reflection ellipsometry (TIRE) and provides better sensitivity compared to SPR [203]. Using TIRE, the OTA aptasensor has been reported. The DNA aptamers were chemisorbed on a gold layer of a glass slide. TIRE allowed the highly sensitive detection of OTA with minimal detectable concentration of 0.01 ng/mL [204]. Most recently, the detection of zearalenone with a LOD of 0.08 ng/mL and detection range of 0.01–1000 ng/mL using the TIRE method has also been reported [205]. The aptasensor was tested on spiked cereal samples, with recoveries of 95.2–104.7%. Sample treatment involved sample grounding and extraction of the analyte with aqueous methanol.

Thus, SPR based aptasensors revealed rather high sensitivity comparable with those of electrochemical sensors. Thus, approximately identical LOD of OTA (0.01 ng/mL) has been reported for electrochemical sensors with DPV signal measurement [168] and a TIRE based biosensor [204].

## 6. Conclusions

Aptamers offer unique opportunities for assembling biosensors intended for the fast and reliable detection of many biologically active species. The interest in mycotoxins detection is mostly related to the extremely low levels of their toxic concentrations and difficulties in their timely detection in agriculture and food safety control. High efficiency of aptamer selection and the specificity of target interactions make it possible to avoid labor- and time-consuming sample treatment. In many cases, a matrix effect can be eliminated by the dilution of the sample or extract. As a result, such aptasensors can be used in field as "point-on-demand" devices. They can be used easily by labor staff and to not have any high use requirements. This makes them competitive against universal conventional techniques, e.g., HPLC or capillary electrophoresis.

Considering the aptasensors described (see Table 4 and below), it should be mentioned that their figures of merit are quite different from each other even though similar aptamers and detection principles were used. This can be attributed to different approaches to the metrological assessment of the measurement results. Thus, most attention was paid to the detection of dangerous levels of mycotoxins but not to the determination of their definite content in the sample. Switch-on/switch-off biosensors show a rather narrow linear range of determined concentrations, and this leaves enough space for diverse interpretation of the results.

Only a few articles offer a direct comparison of the analysis results obtained with aptasensor and HPLC, so that reliability of the real sample assay should be further proved in a more convincing way. Some of the aptasensors describe the simultaneous determination of several mycotoxins. The further progress in such a multiplexed assay can be expected from the extension of the number of labels that the aptamer supports used. Increasing interest to the application of MOFs [168,206] and carbon nanoparticles [135,145,155,179, 207,208] is an example of such trends.

Regarding the comparison of electrochemical and SPR sensors, the use of voltammetric techniques has the advantages of lower cost and higher result sensitivity. On the other hand, electrochemically active species like oxidizable organic species in plant tissues, milk, and wine can interfere with mycotoxin response. Portable SPR instruments remain rather exotic but are less sensitive to the reactions on the chip interface [209]. Contrary to many other optical assay approaches, the target interactions (aptamer–mycotoxin) are realized on the side of the chip opposite to laser beam propagation. This means that the SPR detection is free from the common limitations of optical methods related to bleaching colored labels or chemical reactions induced by irradiation. Being universal, SPR aptasensors for mycotoxin determination can be effectively adapted to other small molecules detection.

1. Although the review covers only recent publications in the area of aptasensors for mycotoxin determination, some trends in further progress can be mentioned;
2. Extension of the nature and characteristics of the labels applied for signal generation is expected, especially in semi-quantitative analysis and "point-on-demand" applications [14,210];
3. More attention will be paid to the combination of aptasensing with microfluidics and simplified biosensor formats, e.g., paper-based biosensors [211] and colorimetric devices based on SERS principles;
4. The focus on the development of new measurement formats will be shifted to the signal-on (switch-on) aptasensors offering better metrological characteristics, especially in real sample assay;
5. The interest in the selection of new aptamer structures and their derivatization in favor of aptasensor assembling will improve both the operational and analytical characteristics of aptasensors and result in the formation of chimeric materials combining aptasensing with the artificial 3D structures of synthetic materials;
6. In general, further efforts in aptasensor design will extend to the area of monitoring the environment and foodstuffs to establish safer and more comfortable life for the population.

**Author Contributions:** Conceptualization, T.H. and G.E.; methodology, A.P. and T.K.; writing—original draft preparation, T.H. and G.E.; writing—review and editing, G.E. and T.H. All authors have read and agreed to the published version of the manuscript.

**Funding:** T.K. acknowledges funding by the subsidy allocated to Kazan Federal University for the state assignment in the sphere of scientific activities (grant No. 0671-2020-0063). T.H. acknowledges funding from the Science Grant Agency VEGA, project No.: 1/0419/20.

**Institutional Review Board Statement:** Not applicable.

**Informed Consent Statement:** Not applicable.

**Data Availability Statement:** Not applicable.

**Conflicts of Interest:** The authors declare no conflict of interest.

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
