# Peer review of "Recent Achievements in Electrochemical and Surface Plasmon Resonance Aptasensors for Mycotoxins Detection"

_chemosensors, doi:10.3390/chemosensors9070180_

Round 1

Reviewer 1 Report

"Biochemical sensors (biosensors) differ from chemical sensors mainly based on traditional detection principles by involvement of biochemical interactions mimicking adverse effects of analytes on human beings [17]. For this purpose, they include biochemical components (enzymes, antibodies, nucleic acids etc.) near the transducer interface so that target interactions affect physical-chemical properties transformed into the electric signal. Biosensors show unique sensitivity and selectivity toward many contaminants exerting acute toxicity [18]."

I do not agree with the notion that biosensors need to have a correlation to toxicity or biological function. Antibodies have a completely different function in the body as in a biosensor. Aptamers are synthetic constructs anyway.

Several typos in Fig. 1: Afatoxin -> Aflatoxin

l. 239 + 244: affine chromatography -> affinity chromatography

I do not agree that this approach is considered to be an affinity chromatographic step. 

Table 2. I do not agree with nearly any of these statements. Most likely, the opposite is true. Although the citation is correct, these wrong statements should not be propagated.

Fig. 5 seems to be wrong in several aspects.

l. 529: "SPR can be applied for detection of sufficiently large molecules with molecular weight above 10 kDa." This comment is wrong or at least misleading. 

l. 546: "The sandwich assay is two step process." This is a strange statement in the context of an SPR assay.

Conclusions: These comments are out of context and do not reflect the rest of the paper. Many of the disadvantages might be relevant, but suitable citations or arguments need to be shown.

In general, the paper contains significant sections of content, which I would bluntly define as wrong. The different sections are inconsistent. I am sorry, but I do not see a way to revise the paper in a sensible way.

Author Response

We are grateful to this reviewer for most useful and stimulating comments that allowed us to improve manuscript. The detailed response to the comments is below. They have been considered in the revised manuscript.

Comment: "Biochemical sensors (biosensors) differ from chemical sensors mainly based on traditional detection principles by involvement of biochemical interactions mimicking adverse effects of analytes on human beings [17]. For this purpose, they include biochemical components (enzymes, antibodies, nucleic acids etc.) near the transducer interface so that target interactions affect physical-chemical properties transformed into the electric signal. Biosensors show unique sensitivity and selectivity toward many contaminants exerting acute toxicity [18]." I do not agree with the notion that biosensors need to have a correlation to toxicity or biological function. Antibodies have a completely different function in the body as in a biosensor. Aptamers are synthetic constructs anyway.

Response: The following text has been introduced instead of the paragraph cited:

"Biochemical sensors (biosensors) utilize specific interactions between the analyte molecules and some native biochemicals, many of which mimic real metabolic paths and adverse effects of analytes on human beings [17]. In biosensor assembly, enzymes, antibodies, nucleic acids etc. are attached near the transducer interface so that target interactions affect physical-chemical properties transformed into the physical signal."

Comment: Several typos in Fig. 1: Afatoxin -> Aflatoxin

Response: the misprints were corrected.

Comment: l. 239 + 244: affine chromatography -> affinity chromatography

Response: the terms „affine immobilization“ and affine chromatography have been substituted by „affinity immobilization“ and „affinity chromatography“trough the text.

Comment: I do not agree that this approach is considered to be an affinity chromatographic step. 

Response: Regarding application of affinity chromatography columns, we can refer to many articles and reviews where the SELEX protocol is linked to some variations of affinity chromatography separation. As one of examples, a short citation from the abstract of the review of E. Kowalska, F. Bartnicki, K. Pels, and W. Strzalka „The impact of immobilized metal affinity chromatography (IMAC) resins on DNA aptamer selection“ (Anal Bioanal Chem. 2014; 406(22): 5495–5499): „The commonly used aptamer selection procedure is systematic evolution of ligands by exponential enrichment (SELEX) where the target molecule is immobilized on an appropriate chromatography resin. For peptide/protein targets, immobilized metal affinity chromatography (IMAC) resins are frequently used“. Nevertheless, to avoid confusing, we have extended the phrase in revised manuscript as follows (page 7 of revised manuscript):

"As was mentioned in the Introduction, aptamers are artificial oligonucleotides selected by combinatorial chemistry from a random library of appropriate nucleotides and separated against an analyte by means of affinity chromatography columns, nitrocellulose binding assay filters or magnetic beads."

Comment: Table 2. I do not agree with nearly any of these statements. Most likely, the opposite is true. Although the citation is correct, these wrong statements should not be propagated.

Response: We believe there is some misunderstanding with the statements. In particularly Reviewer 2 was satisfied with the characteristics done in the Table 2. However, in order to avoid further discussions and confusions, we decided to remove Table 2. Instead, we have added the following paragraph (pages 9-10 of the revised manuscript):

"Regarding the comparison of antibodies and aptamers, higher thermal and chemical stability are mostly mentioned for aptamers utilized in appropriate biosensors [109]. Although the use of nanobodies [131] and Fab antibody fragments [132] compensates for the difference in the molecular weight and size of the receptors. Site specific immobilization of aptamers offers higher density of their positioning on the support and higher sensitivity of appropriate signals toward an analyte binding. Direct comparison of both receptors has been presented for optical detection of analyte binding in [133]. Nevertheless, in some works devoted to sandwich immunoassay, aptamers were utilized instead of primary/secondary antibody because of a less steric hindrance of binding and higher sensitivity of label detection. Thus, in cardiac infarction biomarker detection, the use of aptamer increased twice the current response toward the analyte in picomolar range of its concentration [134]. In sulfamethazine detection, signal enhancement was related to the accumulation of redox indicator, Methylene blue (MB), after the formation of antibody-analyte-aptamer complex on the surface of electrode modified with carboxylated g-C3N4 layer [135]. In the future, similar approach might be interesting in mycotoxin determination, too."

Comment: Fig. 5 seems to be wrong in several aspects.

Response: Probably, more explanation would clarify the Reviewer’s meaning because the reactions presented in Fig. 5 are rather trivial. Thus, the reaction of glutaraldehyde with primary amino group is well documented as glutaraldehyde cross-linking. In some cases, it is accompanied with the following reduction of the Schiff base formed but this is not necessary and is rarely used in biosensor assembling. The couple EDC-NHS as a canonical reactant mixture for binding species with terminal carboxylic group and primary amino group. There might be some alternatives in the order and necessity of single steps separated in the Figure but the result is the same, i.e., the formation of amide bond and covalent binding of the aminated aptamer to carboxylated carrier. Regarding third way, aryl diazonium salt based immobilization protocol become popular in the last decade and in some cases involves electrochemical reduction of the nitro group (electrogrtafting). For this reason, the term „reduction“ was chosen instead of electrons/hydrogen ions transferred. All three protocols mentioned were also used in real aptasensor descriptions summarized below in Table 4.

Comment: l. 529: "SPR can be applied for detection of sufficiently large molecules with molecular weight above 10 kDa." This comment is wrong or at least misleading.

Response: The estimate was taken from Ref. [195,196]. Indeed, application of the method for detection of small molecules is mostly based on some amplification approaches including labeling, structure rearrangements, formation of specific complexes etc. The following sentence is proposed to avoid misleading mentioned (page 17 of the revised manuscript):

"SPR is traditionally applied for detection of sufficiently large molecules with molecular weight above 10 kDa that affect the dielectric properties of the interface [195]. Application of the method for small molecules assay requires amplification strategies, e.g., labeling analytes with appropriate species including nanoparticles, their involvement in the complexes, sandwich or competitive inhibition assay [196]."

Comment: l. 546: "The sandwich assay is two step process." This is a strange statement in the context of an SPR assay.

Response: Sandwhich assay always assumes the use of two receptors that interact with an analyte molecule (form „sandwich“). This description does not depend on the method used for detection of such interactions. Here, two-step SPR assay is described and there is nothing strange. Maybe, the esteemed Reviewer decided sandwich protocol is not optimal for SPR detection scheme commonly applied in kinetic measurement of biochemical interactions, but an example of such SPR aptasensor was provided after the phrase cited.

Comment: Conclusions: These comments are out of context and do not reflect the rest of the paper. Many of the disadvantages might be relevant, but suitable citations or arguments need to be shown.

Response: We have removed the sentences which were assessed as „out of context“. The beginning of the conclusion section was re-written as follows:

"Aptamers offers unique opportunities for assembling biosensors intended to the fast and reliable detection of many biologically active species. The interest to mycotoxins detection is mostly related to extremely low level of their toxic concentrations and difficulties of their timely detection in agriculture and food safety control. High efficiency of aptamer selection and specificity of target interactions make it possible to avoid labor- and time-consuming sample treatment. In many cases, matrix effect can be eliminated by dilution of the sample or extract. As a result, such aptasensors can be used in field as “point-on-demand” devices. They do not mostly offer high requirements to the labor staff. This makes them competitive against universal conventional techniques, e.g., HPLC or capillary electrophoresis.

          Considering the aptasensors described (see Table 4 and below), it should be mentioned that their figures of merit are quite different from each other though similar aptamers and detection principles were used. This can be attributed to different approaches to the metrological assessment of the measurement results. Thus, most attention was paid to the detection of dangerous levels of mycotoxins but not to the determination of their definite content in the sample. Switch-on/switch-off biosensors show rather narrow linear range of concentrations determined and this leaves enough space for diverse interpretation of the results.

          Only few articles offer direct comparison of the analysis results obtained with aptasensor and HPLC so that the reliability of the real sample assay should be further proved in a more convincing way. Some of the aptasensors describe simultaneous determination of a number of mycotoxins. The further progress in such multiplexed assay can be expected from extension of the number of labels of aptamer supports used. Increasing interest to the application of MOFs [168,206] and carbon nanoparticles [135,145,155,179,209,209] is an example of such trends."

Second part of Conclusion was also extended in accordance with other reviewers’ requirements.

Comment: In general, the paper contains significant sections of content, which I would bluntly define as wrong. The different sections are inconsistent. I am sorry, but I do not see a way to revise the paper in a sensible way.

Response: It is difficult to reply to such a general statement, but we would like to draw attention of esteemed Reviewer to the changes made in the revised text, re-written Conclusion and extended SPR aptasensor description, which were made in revised version. All concrete comments of reviewer have been taking into account in the revised manuscript. We hope that carefully revised manuscript and detailed response to the reviewer's comments can be acceptable for this expert.

Reviewer 2 Report

This manuscript reviews recent advances in electrochemical and SPR aptasensors for the detection of mycotoxins. I think that this is well organized, with a general introduction, various assay formats, and their performance factors. It is recommended to accept this journal after supplementing the following comments.

1. The characteristics of aptamers are compared to those of antibodies in Table 2, but a more direct comparison should be made when the two receptors were used for electrochemical and SPR detection.

2. Please discuss Pros and Cons between the electrochemical and SPR aptasensors

3. The authors need to describe future trends in this field, which include aspects in terms of point-of-care testing.

Author Response

This manuscript reviews recent advances in electrochemical and SPR aptasensors for the detection of mycotoxins. I think that this is well organized, with a general introduction; various assay formats, and their performance factors. It is recommended to accept this journal after supplementing the following comments.

Response: We are grateful to this reviewer for positive opinion and for most useful and stimulating comments that allowed us to improve manuscript. The detailed response to the comments is below. They have been considered in the revised manuscript.

Comment 1: The characteristics of aptamers are compared to those of antibodies in Table 2, but a more direct comparison should be made when the two receptors were used for electrochemical and SPR detection.

Response: Unfortunately, we have removed Table 2 in accordance with the requirement of the Reviewer 1, who considered it as those containing mostly wrong statements. We were not agree with such a meaning but decided to remove the Table to avoid further discussion. Instead, we have added the following paragraph (pages 9-10 of revised manuscript):

"Regarding the comparison of antibodies and aptamers, higher thermal and chemical stability are mostly mentioned for aptamers utilized in appropriate biosensors [109]. Although the use of nanobodies [131] and Fab antibody fragments [132] compensates for the difference in the molecular weight and size weight of the receptors, site specific immobilization of aptamers offers higher density of their positioning on the support and higher sensitivity of appropriate signals toward an analyte binding. Direct comparison of both receptors has been presented for optical detection of analyte binding in [133]. Nevertheless, in some works devoted to sandwich immunoassay, aptamers were utilized instead of primary/secondary antibody because of a less steric hindrance of binding and higher sensitivity of label detection. Thus, in cardiac infarction biomarker detection, the use of aptamer increased twice the current response toward the analyte in picomolar range of its concentration [134]. In sulfamethazine detection, signal enhancement was related to the accumulation of redox indicator, Methylene blue (MB) after the formation of antibody-analyte-aptamer complex on the surface of electrode modified with carboxylated g-C3N4 layer [135]. In the future, similar approach might be interesting in mycotoxin determination, too."

Comment 2. Please discuss Pros and Cons between the electrochemical and SPR aptasensors

Response: The following sentences were added to the Conclusion section:

"Regarding the comparison of electrochemical and SPR sensors, the use of voltammetric techniques has advantages of lower cost and higher sensitivity of the results. On the other hand, electrochemically active species like oxidizable organic species in plant tissues, milk and wine can interfere with mycotoxin response. Portable SPR instrumentations remain rather exotic but are less sensitive to the reactions on the chip interface. Contrary to many other optical assay approaches, the target interactions (aptamer – mycotoxin) are realized on the side of the chip opposite to laser beam propagation. This means that SPR detection is free from common limitations of optical methods related to bleaching colored labels or chemical reactions induced by irradiation. Being universal, SPR aptasensors for mycotoxin determination can be effectively adapted to other small molecules detection."

Comment 3. The authors need to describe future trends in this field, which include aspects in terms of point-of-care testing.

Response: Mycotoxin determination is mostly used in foodstuffs and agriculture assay, so that we suggest to use a similar term „point-on-demand“ instead of „point-of-care“ more suitable in medicine. The following text was added into the Conclusion:

"Aptamers offers unique opportunities for assembling biosensors intended to the fast and reliable detection of many biologically active species. The interest to mycotoxins detection is mostly related to extremely low level of their toxic concentrations and difficulties of their timely detection in agriculture and food safety control. High efficiency of aptamer selection and specificity of target interactions make it possible to avoid labor- and time-consuming sample treatment. In many cases, matrix effect can be eliminated by dilution of the sample or extract. As a result, such aptasensors can be used in field as “point-on-demand” devices. They do not mostly offer high requirements to the labor staff. This makes them competitive against universal conventional techniques, e.g., HPLC or capillary electrophoresis."

And below in the same section:

"Although the review covers only recent publications in the area of aptasensors for mycotoxin determination, some trends in the further progress can be mentioned.

  1. Extension of the nature and characteristics of labels applied for signal generation is expected, especially in semi-quantitative analysis and “point-on-demand” applications [14,211].
  2. More attention will be paid to the combination of aptasensing with microfluidics and simplified biosensor formats, e.g., paper-based biosensors [212] and colorimetric devices based on SERS principles.
  3. The focus in the development of new measurement formats will be shifted to the signal-on (switch-on) aptasensors offering better metrological characteristics, especially in real sample assay.
  4. The interest to selection of new aptamer structures and their derivatization in favor of aptasensor assembling will improve both operational and analytical characteristics of aptasensors and result in formation of chimeric materials combining aptasensing with artificial 3D structures of synthetic materials.

In general, further efforts in aptasensor design will extend the area of monitoring of environment and foodstuffs to establish safer and more comfortable life for population."

Reviewer 3 Report

The work reviews recent achievements in electrochemical and SPR aptasensors for detection of mycotoxins. Although the group has published a relevant review in Chemosensors (Electrochemical immuno- and aptasensors for mycotoxin determination, Chemosensors 2019, 7, 10) here they focus on the findings presented in the literature during the last three years regarding electrochemical and SPR aptasensors. Therefore, I consider it has enough novelty and will be useful for the readers dealing with aptasensing. It is written in a clear and concise manner and I suggest its acceptance after minor revision.

Below I summarize my comments.    

Line 102. “Some properties of most common mycotoxins are listed in Table 1.”  The Table does not list properties of the mycotoxins. Please, revise accordingly.

Fig 1. Correct the names depicted under all aflatoxins chemical structures by replacing “Afa…” with “Afla…”

Lines 583-584. “most aptasensors with no respect of еру signal measurement mode belong to the group of switch-off . Please rephrase. It is difficult to understand what it means. (What does “еру” mean?)

Reference 105 is the same with ref. 116

Reference 117 is the same with ref. 146.

In addition, there are several typos throughout the manuscript that should be corrected after careful reading. Below some of them are listed.  

Line 21. Comma (instead of full stop) after the word “concentrations”

Line 54. Replace “they” with “their”

Line 64. Replace… “with the of antigen-antibody” with “with that of antigen-antibody”

Line 81. Full stop.

Line 100. “serious” instead of “serios”  

Line 187. (LC-MS)

Line 244. “affinity chromatography” (affine has a different meaning)

Line 612. “semi-quantitative”

Line 624. Replace “comfortable lie” with “comfortable life”

Line 875. “aptasensors”

Author Response

The work reviews recent achievements in electrochemical and SPR aptasensors for detection of mycotoxins. Although the group has published a relevant review in Chemosensors (Electrochemical immuno- and aptasensors for mycotoxin determination, Chemosensors 2019, 7, 10) here they focus on the findings presented in the literature during the last three years regarding electrochemical and SPR aptasensors. Therefore, I consider it has enough novelty and will be useful for the readers dealing with aptasensing. It is written in a clear and concise manner and I suggest its acceptance after minor revision. Below I summarize my comments.

Response: We are grateful to this reviewer for positive opinion and for most useful comments that allowed us to improve manuscript. The detailed response to the comments is below. They have been considered in the revised manuscript.

Comment: Line 102. “Some properties of most common mycotoxins are listed in Table 1.”  The Table does not list properties of the mycotoxins. Please, revise accordingly.

Response: The phrase was changed as follows: „Source, potential danger and maximal admissible levels of common mycotoxins are listed in Table 1.“

Comment: Fig 1. Correct the names depicted under all aflatoxins chemical structures by replacing “Afa…” with “Afla…”

Response: We thank to esteem Reviewer for notification, misprints were corrected.

Comment: Lines 583-584. “most aptasensors with no respect of еру signal measurement mode belong to the group of switch-off. Please rephrase. It is difficult to understand what it means. (What does “еру” mean?)

Response: Actually, this was Russian „any“ – we are sorry for the sudden switching language. We have changed the first part of Conclusion in accordance with the requirement of the Reviewer 1 as follows:

"Aptamers offers unique opportunities for assembling biosensors intended to the fast and reliable detection of many biologically active species. The interest to mycotoxins detection is mostly related to extremely low level of their toxic concentrations and difficulties of their timely detection in agriculture and food safety control. High efficiency of aptamer selection and specificity of target interactions make it possible to avoid labor- and time-consuming sample treatment. In many cases, matrix effect can be eliminated by dilution of the sample or extract. As a result, such aptasensors can be used in field as “point-on-demand” devices. They do not mostly offer high requirements to the labor staff. This makes them competitive against universal conventional techniques, e.g., HPLC or capillary electrophoresis.

          Considering the aptasensors described (see Table 4 and below), it should be mentioned that their figures of merit are quite different from each other though similar aptamers and detection principles were used. This can be attributed to different approaches to the metrological assessment of the measurement results. Thus, most attention was paid to the detection of dangerous levels of mycotoxins but not to the determination of their definite content in the sample. Switch-on/switch-off biosensors show rather narrow linear range of concentrations determined and this leaves enough space for diverse interpretation of the results.

          Only few articles offer direct comparison of the analysis results obtained with aptasensor and HPLC so that the reliability of the real sample assay should be further proved in a more convincing way. Some of the aptasensors describe simultaneous determination of a number of mycotoxins. The further progress in such multiplexed assay can be expected from extension of the number of labels of aptamer supports used. Increasing interest to the application of MOFs [168,207] and carbon nanoparticles [135,145,155,179,208,209] is an example of such trends."

Comment: Reference 105 is the same with ref. 116. Reference 117 is the same with ref. 146.

Response: We thank to the Reviewer for careful reading of the manuscript and for this notification, the references were re-numbered.

Comment: In addition, there are several typos throughout the manuscript that should be corrected after careful reading. Below some of them are listed.  

Line 21. Comma (instead of full stop) after the word “concentrations”

Line 54. Replace “they” with “their”

Line 64. Replace… “with the of antigen-antibody” with “with that of antigen-antibody”

Line 81. Full stop.

Line 100. “serious” instead of “serios”  

Line 187. (LC-MS)

Line 244. “affinity chromatography” (affine has a different meaning)

Line 612. “semi-quantitative”

Line 624. Replace “comfortable lie” with “comfortable life”

Line 875. “aptasensors”

Response: We thank to esteem Reviewer for notification. The corresponding changes have been made as requested. We have also checked the manuscript again to avoid misprints.

Reviewer 4 Report

The work presented by Evtugyn et al. is a review describing the use of aptasensors in the sensitive determination of mycotoxins, which are secondary metabolites produced by fungi that affect a wide range of agricultural products. Their determination in food and beverages is of great interest since these substances are poisonous to humans if consumed in sufficient quantities. The review is mainly focused on the application of electrochemical and SPR aptasensors for mycotoxin determination in environmental samples and food described in the period from 2018 to 2021.

In the first part of the review, the authors describe the different types of mycotoxins, and then briefly summarize the conventional methods used for their determination. The second part of the article is then focused on describing the use of aptamers as biorecognition elements in de development of biosensors as well as their immobilization strategies. Finally, the authors review the aptasensors developed in recent years for the determination of mycotoxins using electrochemical transducers and optical transducers based on surface plasmon resonance (SPR).

The article is well written. Authors show large experience in the sensor and biosensor field, especially based on DNA aptamers, publishing their works in high-impact journals. Thus, I would consider the paper acceptable for publication after the addressing minor points:

  1. While the review of electrochemical aptasensors is very extensive and detailed, the one on SPR aptasensors is in contrast too short (less than two pages). This makes the title of the article does not reflect its content. I understand that it is difficult for the authors to expand the SPR part of the article when only 4 or 5 aptasensors using this technique for mycotoxin detection have been developed/published to date, but the title and abstract of the article leads one to believe that this is not the case. In this sense, I suggest the authors to reconsider the title of the article.
  2. The recent review: “Aptasensors for mycotoxin detection: A review”   (https://doi.org/10.1016/j.ab.2021.114156) should be referenced in the present work.
  3. The conclusions of the paper are more focused on the detection of mycotoxins using aptasensors, in general, than on the detection of mycotoxins using electrochemical and SPR aptasensors. It is difficult to relate some of the conclusions achieved by the authors to the rest of the manuscript. At no point is there any mention of the two transducers that are the subject of the review.
  4. In my opinion, it is a little hard to read the manuscript. It is maybe a problem of the pdf format, but figures, text and tables have different vertical margins and figure 7 is not complet.

Author Response

The work presented by Evtugyn et al. is a review describing the use of aptasensors in the sensitive determination of mycotoxins, which are secondary metabolites produced by fungi that affect a wide range of agricultural products. Their determination in food and beverages is of great interest since these substances are poisonous to humans if consumed in sufficient quantities. The review is mainly focused on the application of electrochemical and SPR aptasensors for mycotoxin determination in environmental samples and food described in the period from 2018 to 2021.

In the first part of the review, the authors describe the different types of mycotoxins, and then briefly summarize the conventional methods used for their determination. The second part of the article is then focused on describing the use of aptamers as biorecognition elements in de development of biosensors as well as their immobilization strategies. Finally, the authors review the aptasensors developed in recent years for the determination of mycotoxins using electrochemical transducers and optical transducers based on surface plasmon resonance (SPR).

The article is well written. Authors show large experience in the sensor and biosensor field, especially based on DNA aptamers, publishing their works in high-impact journals. Thus, I would consider the paper acceptable for publication after the addressing minor points:

Response: We are grateful to this reviewer for positive opinion and for most useful comments that allowed us to improve manuscript. The detailed response to the comments is below. They have been considered in the revised manuscript.

Comment 1: While the review of electrochemical aptasensors is very extensive and detailed, the one on SPR aptasensors is in contrast too short (less than two pages). This makes the title of the article does not reflect its content. I understand that it is difficult for the authors to expand the SPR part of the article when only 4 or 5 aptasensors using this technique for mycotoxin detection have been developed/published to date, but the title and abstract of the article leads one to believe that this is not the case. In this sense, I suggest the authors to reconsider the title of the article.

Response: We have extended the part related to SPR sensors at Conclusion in respect of their comparison with electrochemical sensors. We hope this will be sufficient to leave both detection principles in the article title.

Comment 2: The recent review: “Aptasensors for mycotoxin detection: A review”   (https://doi.org/10.1016/j.ab.2021.114156) should be referenced in the present work.

Response: The review was added as requested (ref. [26]).

Comment 3: The conclusions of the paper are more focused on the detection of mycotoxins using aptasensors, in general, than on the detection of mycotoxins using electrochemical and SPR aptasensors. It is difficult to relate some of the conclusions achieved by the authors to the rest of the manuscript. At no point is there any mention of the two transducers that are the subject of the review.

Response: We have re-written whole Conclusions.

Comment 4: In my opinion, it is a little hard to read the manuscript. It is maybe a problem of the pdf format, but figures, text and tables have different vertical margins and figure 7 is not complete.

Response: We have checked the format of the pdf file and reformated Fig. 7 as requested.